# Characteristics and Trends of Workplace Violence towards Frontline Health Workers under Comprehensive Interventions in a Chinese Infectious Disease Hospital

**DOI:** 10.3390/healthcare12191911

**Published:** 2024-09-24

**Authors:** Yiming Huang, Min Zhang, Chuning He, Fuyuan Wang, Yujie Liu, Jing Wu, Qianqian Luo, Na Chen, Yuting Tang

**Affiliations:** 1School of Population Medicine and Public Health, Chinese Academy of Medical Sciences & Peking Union Medical College, Beijing 100730, China; huangyiming@sph.pumc.edu.cn (Y.H.); hechuning@student.pumc.edu.cn (C.H.); wangfuyuan@student.pumc.edu.cn (F.W.); jinmlee@163.com (Y.L.); jing.wu296@gmail.com (J.W.); luoqq4677@163.com (Q.L.); tyting0109@student.pumc.edu.cn (Y.T.); 2National Centre for Occupational Safety and Health, National Health Commission, Beijing 102308, China; 17611134532@163.com

**Keywords:** workplace violence, occupational health, health workers, OSHMS, HealthWISE, risk assessment

## Abstract

Objectives: This study investigated workplace violence (WPV) toward frontline health workers under comprehensive interventions to improve the occupational safety and health management system in a Chinese infectious disease hospital. Methods: The risk assessment of WPV using an international questionnaire was conducted in 2018 and 2021 to compare the perceived levels of exposure to WPV and intervention measures before and after the intensification of anti-violence measures in the hospital context. Additionally, qualitative data were collected in 2021 through semi-structured and unstructured interviews, providing complementary information about WPV toward frontline health workers (HWs). Results: After establishing the occupational safety and health management system (OSHMS), the total incidence rate of WPV decreased from 60.90% in 2018 to 34.44% in 2021. Psychological violence declined significantly from 60.90% in 2018 to 33.89% in 2021. The endorsement of precautionary measures increased significantly from 2018 to 2021, including patient screening recognition, patient protocol, shift or rota changes, etc. A thematic analysis of several subthemes shows that HWs had an in-depth understanding of WPV, recognizing its multifaceted consequences in the context of complex risk factors. Conclusions: This study demonstrates a significant decrease in WPV, psychological violence, verbal abuse, bullying/mobbing, and ethnic discrimination after implementing the comprehensive OSHMS.

## 1. Introduction

Workplace violence (WPV) in health settings has been emerging as a global public health concern. Health facilities have been recognized as having a heightened risk of WPV [1], which has adverse effects on victims and creates an atmosphere of insecurity and fear in the workplace [2,3,4]. Post-traumatic stress disorder (PTSD) resulting from violence and harassment was included in the International Labour Organization (ILO) list of occupational diseases (2010) [5]. In the global social-economic context, it is imperative to persist in combatting this pervasive phenomenon, which was further exacerbated by the COVID-19 pandemic [6,7].

*The Violence and Harassment Convention No. 190* [8] and *the Violence and Harassment Recommendation No. 206* [9] were adopted in the Centenary International Labour Conference, marking the first international labour standard to address violence and harassment in the world of work. These instruments provide a shared framework for action and a unique opportunity to shape the future of work founded on principles of social justice. In June 2021, the ILO published a guide titled “*Violence and Harassment in the World of Work: A Guide on Convention No. 190 and Recommendation No. 206*” [10]. The definition of violence and harassment in the world of work encompasses “a range of unacceptable behaviors and practices… [that] aim at, result in, or are likely to result in physical, psychological, sexual, or economic harm”.

In February 2021, the World Health Organization (WHO) and the ILO jointly issued interim guidance titled “*COVID-19: Occupational Health and Safety for Health Workers*”. This guide outlines the most effective measures, including WPV prevention and control, harassment, and discrimination against health workers (HWs) to prevent and mitigate COVID-19 at work [11]. In the five years since the adoption of C190, 44 ratifications have been made, and more are in the pipeline [12]. Notably, China has bolstered its regulations against WPV in alignment with *Convention No. 190*, for instance, *the Law of Peoples Republic of China on Basic Health Care, Medicine, and Health Promotion (adopted in 2019), the Civil Code of the People’s Republic of China* (hereafter referred to as *the China Civil Code*) (adopted in 2020), *the Law of the People’s Republic of China on Doctors (adopted in 2021), and the Law of People’s Republic of China on the Protection of Rights and Interests of Women (adopted in 2005 and secondly revised in 2022)*—the aforementioned laws mandate the protection of HWs’ personal safety, dignity, and legitimate rights and interests against WPV [13,14,15,16,17].

Simultaneously, a substantial body of quantitative studies on WPV in the Chinese health sector has been published. During the last five years, a rapid literature search was conducted in the databases of PubMed with three key words, (workplace violence [Abstract]) and (health worker [Abstract]) and (China [Abstract]), and 108 English articles were found, with the majority being quantitative studies, particularly cross-sectional studies. A limited systematic review/meta-analysis indicated a growing focus on HW safety in China due to numerous incidents of hospital-based violence against medical professionals; the proportion of WPV exposure differed greatly across study locations, practice settings, work schedules, and occupations [18,19,20]. Despite this, only a few qualitative studies were identified, including a qualitative interview on prevention and management strategies for external consequences resulting from sexual harassment in the workplace during the COVID-19 crisis [21], a qualitative comparative analysis exploring the causes of patient violence toward medical staff in China during the COVID-19 pandemic [22], and a qualitative comparative analysis evaluating the judicial judgments and media sensation of violence against medical staff in China [23]. However, it is important to note that the existing body of research is predominantly rooted in the Chinese cultural context, which may not fully capture the characteristics of WPV from an international perspective. Moreover, there is a conspicuous absence of mixed-methods studies that could provide a more comprehensive assessment of the efficacy of intervention strategies against WPV within the framework of the OSHMS in Chinese hospitals, particularly when viewed through an international standpoint.

Technical support plays a pivotal role in developing global and national policies. In 2014, the WHO and ILO jointly initiated the international technical tool known as “HealthWISE-Work Improvement in Health Services” [24,25]. HealthWISE is a practical, participatory methodology for improving the quality of health facilities, based on the principles of the ILO’s programme “Work Improvement in Small Enterprises” (WISE). It promotes the application of smart, simple, and low-cost solutions by utilizing local resources, which leads to tangible benefits for workers and their employers. HealthWISE is designed to promote learning by doing, which encourages managers and staff to work together to raise awareness of occupational safety and health. A green, healthy, family-friendly workplace with high-quality managing equipment and supplies will also be promoted. This, in turn, helps improve health services’ performance and ability to deliver quality care to patients. China has been using HealthWISE to build capacities for work improvement in the health sector by the WHO-ILO-China HealthWISE team (hereafter referred to as the Team). The training prioritized the HealthWISE modules dealing with general control of occupational hazards, musculoskeletal disorders, and biological hazards, and infection control, plus tackling discrimination, harassment, and violence. The adoption of HealthWISE as a sustainable national program in China has been advocated for in more than 260 pilot hospitals [26,27]. Chinese experience and good practices are introduced in the WHO/ILO guide titled “Caring for those who care: guide for the development and implementation of occupational health and safety programmes for health workers” [28].

The hospital in this article has been actively engaged in applying intervention with HealthWISE and national policies/standards since 2015. The key milestones of establishing the OSHMS in the hospital after the HealthWISE TOT Workshop are summarized in Figure 1. It is worth pointing out that the hospital systematically established the Steering Committee of OSH and the OSHMS in 2016, and the Sixth National HealthWISE TOT workshop was held in this hospital in July 2019 after the Centenary International Labour Conference adopted *Convention No. 190* and *Recommendation No. 206*, with the topic of *actively responding to the world of work to eliminate violence and harassment and promote the vision of zero occupational hazards* (Figure 1).

The risk assessment of occupational hazards and improvement of the OSHMS are continuous and interactive over time. The infectious disease hospitals are high-risk workplaces for Occupational Bloodborne Pathogen Exposure (OBPE) and the occupational exposure status of OBPE was first investigated by the Team. In terms of psychosocial hazards, there is a lack of research on WPV in health settings from the perspective of occupational health protection. Considering the accessibility of participants, we conducted two rounds of surveys in 2018 and 2021. It is worth pointing out that the second wave of surveys regarding the risk assessment of WPV was conducted in July 2021, which was a relatively stable period of COVID-19.

This study aims to analyze the characteristics and trends of WPV between the two waves of the survey: (i) by exploring the relationship between sociodemographic and work-related factors and the perceived levels of exposure to (e.g., physical violence, verbal abuse, bullying/mobbing, sexual harassment, and ethnic discrimination); (ii) by comparing the perceived levels of exposure to WPV before and after the intensification of anti-violence measures in the hospital context; and (iii) by describing the perspectives of frontline HWs on exposure to WPV, as well as the relevance of the preventive measures adopted in the hospital context.

## 2. Materials and Methods

### 2.1. Design

We conducted three phases of research in this pilot hospital to verify whether systematically implementing the OSHMS improved occupational health protection among HWs. The first phase of research concentrated on risk assessment of WPV towards HWs. A cross-sectional survey was conducted using an international questionnaire in July 2018 to investigate the occurrence of violence in the past year, defined as the baseline survey. The second phase of the research focused on gathering experiences and lessons for the scale-up application of HealthWISE during the COVID-19 pandemic, demonstrating the systematic improvement of occupational health for HWs by HealthWISE implementation. The corresponding research was published in 2022 by the Team (https://doi.org/10.3389/fpubh.2022.1010059) (accessed on 24 June 2024) [27]. A mixed-methods study was used in the third research phase to investigate the status of WPV among HWs. A cross-sectional survey was conducted using the same questionnaire in July 2021, which was defined as this survey. Simultaneously, semi-structured interviews were conducted to investigate the perceptions of WPV and formulate a strategy for further piloting practice (Figure 1).

### 2.2. Setting

The Hospital is a tertiary hospital with 850 staff and 550 beds, specializing in infectious diseases and comorbidities, particularly, Human Immunodeficiency Virus/Acquired Immunodeficiency Syndrome (HIV/AIDS), Tuberculosis (TB), and Hepatitis. It serves as the designated hospital for COVID-19 patients in the provincial capital city during the COVID-19 pandemic.

### 2.3. Measurement of Quantitative Study

#### 2.3.1. Study Population

The target population included all hospital health workers, i.e., doctors, nurses, medical technician, and administrative staff. The inclusion criteria were as follows: (1) HW with professional certification; (2) voluntary participation in the survey with informed consent; and (3) employed by the hospital as a regular employee for >1 year. Exclusion criteria were as follows: (1) those failing to answer the questionnaire in the opening hours; and (2) those exceeding the time limit for the questionnaire.

In June 2018, valid data from 156 respondents via mobile phone were collected. Participation in the risk assessment of WPV is gradually improved under the systematic implementation of OSHMS. In 2021, the number of HWs who met the inclusion criteria was 820. The data management platform showed that 728 respondents who met the inclusion criteria completed the questionnaire, of whom 720 had valid questionnaires (total response rate 88.78%; and total valid response rate 87.80%).

#### 2.3.2. Questionnaire

Workplace Violence in the Health Sector Country Case Studies Research Instruments—Survey Questionnaire was used. The original language of the questionnaire was English [29]. Before translating it into Chinese, we have formally obtained permission to use the Questionnaire from the ILO. Pretest and retest reliability and validity were conducted by the Team in Beijing and Shenzhen city, with Cronbach’s coefficient α at 0.83.

#### 2.3.3. Data Collection

Members of the Team visited the relevant departments and kept inviting HWs on duty to fill out an online questionnaire using their cell phones in July 2021. The informed consent was presented in the first page of the questionnaire, which could only be accessed by those who had given their informed consent. The Chinese version of the questionnaire is available online (https://www.wjx.cn/vm/OruZfpm.aspx) (accessed on 24 June 2024).

#### 2.3.4. Statistical Analysis

IBM SPSS Statistics V.26.0 (SPSS Inc., Chicago, IL, USA) and Excel (Microsoft Corporation, Redmond, WA, USA) were used: (i) Chi-square test was calculated for the relationship between sociodemographic and work-related factors and the perceived levels of exposure to WPV (e.g., physical violence, verbal abuse, bullying/mobbing, sexual harassment, and ethnic discrimination). Significant factors were modelled in binary logistic regression analysis to calculate ORs with CIs by using the forward stepwise (likelihood ratio) method. The data were examined at a 95% CI, and *p* < 0.05 was deemed statistically significant. The features of occurrence and investigation of WPV among participants were analyzed. Moreover, a comparison of the perceived levels of exposure to WPV before and after the intensification of anti-violence measures in the hospital context was conducted.

#### 2.3.5. Quality Control

All researchers were trained with respect to the background of the investigation. Before the survey, researchers explained the purpose and importance of the study and called for participation from HWs. Two researchers validated the data and deleted logical errors after collecting the questionnaire. The database was then analyzed.

### 2.4. Measurement of Qualitative Research

#### 2.4.1. Interview Data Collection

The semi-structured questionnaire was drafted in both English and Chinese versions which were validated by senior professionals and the Team leader. The finalized questionnaire focused on five themes with ten questions is available in Appendix A. Meanwhile, unstructured interview was conducted with the administrative and security staff through an open discussion on the topic about the measures of WPV intervention in the Hospital, providing complementary information about WPV toward frontline HWs. Considering gender, department, and position relevant to PWV prevention and control, we adopted purposive sampling to select the study participants, and data collection was continued until the saturation point was reached.

The interviews were held face-to-face by the Team members in July, 2021. Informed consent forms for participation and audio-recording were obtained before commencing the interviews. At the start of each interview, the interviewees were briefed on the study background, research purpose, and definitions of relevant terms. During the interview, the Team members established an equal and trusting relationship with the interviewees. Interviews were conducted individually in a quiet setting at the workplace to encourage participants to share their opinions confidently. On average, each interview lasted 30 min. The interviewers and study participants have had no contact up to this point other than to arrange the interviews.

#### 2.4.2. Interviewee Characteristics

A total of 28 interviewees (19 females and 9 males) participated in the study, including 21 frontline HWs (doctors and nurses) for the semi-structured interviews among 13 departments (Intensive Care Unit, Hepatology, Internal Medicine, Surgery, Emergency, Vaccination, TB, Pre-examination and Triage, Ultrasound, Obstetrics and Gynaecology, HIV/AIDS, Vaccination, and General Outpatient), and seven administrative and security staff among eight departments (Trade Union, Nursing Department, Logistics, HIV/AIDS, Infectious Diseases, Security Section, and Security Team) for the unstructured interviews. Afterward, records of the 28 interviewees were filed anonymously under numbers P1–P28 to protect their privacy.

#### 2.4.3. Data Coding and Analysis

All interview recordings were transcribed verbatim in Chinese. The consolidated criteria for reporting qualitative studies [30] are reported in Appendix A. The transcripts were reviewed by the first author to verify the correct transcription. Six-phase thematic analysis of the semi-structured interview data in Chinese was undertaken using Microsoft Excel (Microsoft Corporation, Redmond, WA, USA) and NVivo version 12.0 (QSR International, Doncaster, Australia) [31,32].

## 3. Results

### 3.1. Results of Quantitative Research

#### 3.1.1. Incidence and Distribution of WPV in 2021

Most of the respondents who completed valid questionnaires were women (74.03%) and doctors (47.22%). A total of 24 (3.33%) respondents had encountered physical violence, while 244 (33.89%) had experienced psychological violence in the past 12 months. Verbal abuse (32.63%) was the most common type of psychological violence, followed by bullying/mobbing (8.47%), sexual harassment (2.22%), and ethnic discrimination (1.25%) (Appendix A).

#### 3.1.2. Influencing Factors of WPV in 2021

Table 1 shows the results of binary logistic regression. It indicates that night work is related to the occurrence of physical violence. Respondents in night work were 3.43 times (95% *CI* 1.01 to 11.63) more exposed to physical violence. Professional title, having direct physical contact/interaction with patients, and worrying about WPV are related to the occurrence of verbal abuse. Having direct physical contact/interaction with patients (OR = 6.33, 95% *CI* 2.66 to 15.05) and worrying about WPV (OR = 2.87, 95% *CI* 1.68 to 4.89) increased the risk of experiencing verbal abuse. Professional title, department, and worrying about WPV are related to the occurrence of bullying/mobbing. Respondents with a middle level of professional titles were 2.15 times (95% *CI* 1.14 to 4.07) and those with a senior level of professional titles were 3.09 times (95% *CI* 1.53 to 6.21) more likely to have suffered from bullying/mobbing. Respondents in outpatient and emergency departments were 2.79 times (95% *CI* 1.31 to 5.98) more exposed to bullying/mobbing. Those who worry about WPV were 2.96 times (95% *CI* 1.02 to 8.30) more likely to have suffered from bullying/mobbing.

#### 3.1.3. Features of Occurrence to WPV among Participants in 2021

Among the 24 respondents who had experienced physical violence, most of the perpetrators were patients themselves (41.67%) or the patients’ relatives (37.50%). Most of the physical violence occurred in the hospital (79.17%). Among the respondents who had experienced verbal abuse, bullying/mobbing, sexual harassment, and ethnic discrimination, most of the perpetrators were patients themselves or the patients’ relatives, and most of these occurrences were happened in the hospital.

For the types of perpetrators, patient/client, relatives of patient/client, external colleague/worker, and general public belong to the category of external violence, while staff member and management/supervisor belong to the category of internal violence. For the above types of WPV, the number of perpetrators of external violence was higher than that of internal violence. For internal violence, the proportion of perpetrators who commit sexual harassment is more apparent. Respondents reported having been sexually harassed by staff member (12.50%) or management/supervisor (6.25%) in the last 12 months (Table 2).

#### 3.1.4. Changes in Frequency of WPV between the Two Waves of Survey

For gender, using the Cochran–Armitage (CA) trend test, the *p* value of the trend test is < 0.01 (χ^2^ trend = 6.026), indicating that the change in gender has a trend statistical significance between the two waves of survey, with the proportion of males increasing year by year and the number of females decreasing year by year. For occupation, using the Cochran–Armitage (CA) trend test, the *p* value of the trend test is 0.274 (χ^2^ trend = 1.195), indicating that the change in occupation over time is not statistically significant for the trend (Table 3).

Compared with the baseline survey, the total incidence rate of WPV decreased from 60.90% in 2018 to 34.44% in 2021 (χ^2^ = 37.661, *p* < 0.01). Psychological violence declined significantly from 60.90% in 2018 to 33.89% in 2021 (χ^2^ = 39.427, *p* < 0.01) (Figure 2a). Furthermore, for psychological violence, verbal abuse fell significantly from 60.26% in 2018 to 32.64% in 2021 (χ^2^ = 41.701, *p* < 0.01), while bullying/mobbing dropped significantly from 19.23% in 2018 to 8.47% in 2021 (χ^2^ = 15.943, *p* < 0.01), and ethnic discrimination decreased from 3.85% in 2018 to 1.25% in 2021 (χ^2^ = 5.135, *p* = 0.02) (Figure 2b).

#### 3.1.5. Changes in Recognition of Perceived Anti-Violence Measures in the Workplace

The results revealed that the percentage of restricting public access is increased from 38.46% in 2018 to 72.92% in 2021 (χ^2^ = 68.615, *p* < 0.001), the percentage of patient screening is increased from 34.62% in 2018 to 59.03% in 2021 (χ^2^ = 30.836, *p* < 0.001), the percentage of recognition for patient protocol is increased from 36.54% to 56.94% (χ^2^ = 21.450, *p* < 0.001), the percentage of recognition of changing shifts or rotas increased to 42.08% in 2021 from 30.13% in 2018 (χ^2^ = 7.639, *p* = 0.006), the percentage of recognition for restricting the exchange of money in the workplace is increased from 30.77% to 46.67% (χ^2^ = 13.162, *p* < 0.001), the percentage of recognition for special equipment or clothing is increased from 28.21% to 41.81% (χ^2^ = 9.935, *p* = 0.002), the percentage of recognition for the check-in procedures for staff is increased from 17.95% to 34.72% (χ^2^ = 9.935, *p* = 0.002), and the percentage of recognition for increasing staff numbers is increased from 32.69% to 43.47% (χ^2^ = 6.135, *p* = 0.013) (Figure 3).

### 3.2. Results of Qualitative Research

Five broad themes have emerged from the interviews of frontline HWs, and administrative and security staff. Subthemes and codes derived from each theme have been organized. Verbatim quotes for each theme, subthemes, and codes are provided in Appendix A.

#### 3.2.1. Definition and Experience of WPV Incidents among Frontline HWs

Regarding their understanding of the PWV definition, the majority of interviewees demonstrated a clear understanding of the term (17 out of 21 interviewees). They identified physical violence as encompassing various forms, such as physical harm, physical altercations, personal attacks, threats to life safety, and even incidents of homicide. Psychological violence was identified as involving psychological distrust and rejection, verbal abuse, insults, humiliation, verbal attacks, and verbal threats. Additionally, three interviewees acknowledged that WPV could occur not only between HWs and patients/family member but also among colleagues in the Hospital. In terms of their experience with WPV, 14 from 21 frontline HWs mentioned experiencing WPV in the previous year, which were in the form of verbal abuse from patients and their family members. Additionally, five interviewees mentioned the witnessing or hearing about colleagues facing violence from patients and their family members. The main forms of violence were verbal abuse, intimidation, and sexual harassment.

#### 3.2.2. Factors Linking to Consequences and Causes of WPV among Frontline HWs

The consequences of WPV on HWs can be summarized as follows: (a) Negative impacts on individual (15 out of 21): The majority of interviewees stated that WPV could cause physical harm, leading to psychological pressure. Some interviewees felt a sense of powerlessness/emotional distress during their work. Additionally, WPV could damage the self-esteem of HWs and create an unsafe feeling during their work with the fear of experiencing violence again. However, two interviewees stated that WPV did not have adverse effects on them, and one tried to understand the difficulties of patients instead of dwelling on the violent incidents. (b) Negative impacts on the work organization (7 out of 21): WPV could affect the motivation of health workers, leading to the feelings of apathy. Some interviewees questioned their career choices, and then considered turnover. WPV could also increase the likelihood of work errors. (c) Negative impacts on the family (2 out of 21): HWs experiencing WPV might have negative emotions that affect family harmony. Family members would raise the concerns about the safety of their loved ones with fear. The children might avoid choosing their career in the health sector due to the parents’ experience of WPV.

The causes of physical violence which were reported by frontline HWs can be summarized as follows: (a) Unmet treatment expectation (14 out of 21): Long-term treatment of chronic diseases come with a high cost. There was distrust in the medical practice due to patients’ excessive expectation of treatment outcomes. Both medical staff and patients lacked self-restraint in dealing with conflicts, leading to ineffective or insufficient communication. (b) Perpetrators’ factors (13 out of 21): Perpetrators were under the influence of drugs or alcohol; they had a low education level; they had an irritable personality; they suffered from mental disorder, or loss of control due to antiviral medication; and they were excessive anxious as the parents of child patients. (c) Social factors (4 out of 21): There is often tension in the doctor–patient relationship, and inadequate media coverage of punitive measures against perpetrators in severe cases of violence. (d) Workplace factors (3 out of 21): There was understaffing; tight working schedules; unreasonable long waiting times; restrictions on family accompanying during the COVID-19 pandemic; and inadequate professional training and guidance for dealing with WPV. (e) Victims’ factors (2 out of 21): The HWs were young; they had limited work experience; they exhibited poor service attitudes; or they lacked the ability to effectively handle violent incidents.

The causes of psychological violence from the opinions of frontline HWs can be summarized as follows: (a) Perpetrator’s factors (9 out of 21): Personality traits (impatience, extremism, bad temper); non-compliance with medical procedures; lack of understanding of appointment registration and medical processes; low education level; high economic and psychological pressure; alcohol abuse; and poor mental state. (b) Unmet treatment expectations (6 out of 21): No improvement in the patient’s condition; an information gap between doctors and patients; patients’ poor medical knowledge and misconceptions about the disease; and discrepancies between expectations and reality. (c) Workplace factors (4 out of 21): Unclear or unreasonable processes for medical appointment and service payment; long waiting times for patients with urgent conditions; dissatisfaction with medical procedures; understaffing; and inadequate communication channels. (d) Victim’s factors (3 out of 21): Using a harsh tone during peak work time; being unable to meet patients’ needs promptly; poor communication tone and attitude among colleagues; and young staff being unskilled during operations. (e) Social factors (2 out of 21): Inadequate legislation and measures to address violence against HWs; insufficient punishment for perpetrators; unfriendly policies toward HWs during the reform and development of healthcare system; underinvestment in the healthcare system; and overburden for patient payment.

#### 3.2.3. Measures to Tackle WPV for Frontline HWs

The current measures taken by the Hospital to tackle WPV were reflected in the following aspects: (a) Environmental intervention (13 out of 21): Electronic security scanners were installed at hospital entrances to prohibit controlled knives and dangerous items; one-click alarm devices were equipped in key areas such as outpatient halls, examination rooms, wards, and corridors; separate pathways for HWs were renewed from those for patients and family members; security staff were assigned to each department, and 24 h security positions were set in key departments. (b) Organizational intervention (10 out of 21): The Hospital developed emergency response plans to address WPV with multi-departmental participation, and trained HWs on the reporting process for WPV; a medical dispute office was established with experienced staff; a well-defined operating procedure for patients was taken, such as volunteer assistance, and optimizing the patient treatment process; and an effective mechanism of information exchange and communication was established between medical staff and patients. (c) Individual intervention (5 out of 21): HWs participated in regular training on WPV, strengthening their abilities to identify WPV risks and improve communication skills; and verbal reporting procedures for WPV was clarified by the emergency response plan. (d) Other measures (2 out of 21): The Hospital advocated care and support measures for patients, such as alleviating the patients’ hospitalization expense; and cultural communication posts/bulletins were set up to foster a harmonious doctor–patient relationship.

The effective measures to reduce WPV were noted by the frontline HWs, including the following: (a) Individual-level measures (10 out of 21): Fostering empathy toward patients; and enhancing self-protection with early detection of WPV. (b) Hospital-level measures (9 out of 21): Increasing the number of HWs and security personnel; installing equipment systems for one-touch alarm and security scanners at the entrance; conducting emergency drills to enhance preparedness and response; providing personal accident insurance for emergency medical personnel; and organizing training programs to enhance awareness of communication and service among HWs. (c) Societal-level measures (2 out of 21): Developing health insurance policies to alleviate the economic burden of patients and ease the tension in the doctor–patient relationship.

#### 3.2.4. Approaches and Outcomes of Actions Against WPV during COVID-19

The approaches for addressing WPV during the COVID-19 pandemic were described by the interviewees as follows: (a) Environmental intervention (10 out of 21): Strengthening access control measures for public access; and increasing the number of security personnel and frequency of security patrol. (b) Organizational intervention (7out of 21): Strengthening good practice against WPV; the hospital setting up exemplary HWs/positions; carrying out a flexible work schedule to ensure adequate rest for HWs; and enhancing care and support for HWs with a special focus on psychological issue. (c) Individual-level intervention (1 out of 21): Carrying out specific training programs on WPV.

Five interviewees highlighted the outcomes of the aforementioned actions during the COVID-19 pandemic: (a) They felt much safer during their work time than before; (b) There was a slight improvement in the doctor–patient relationship; and (c) Strengthening security measures during the pandemic had a deterrent effect on emotionally agitated patients.

#### 3.2.5. Plans for Hospital and Individual Improvement of WPV Prevention and Control from Frontline HWs

The interviewees provided the following suggestions of improving measures for the Hospital to address WPV: (a) Strengthening organizational intervention (6 out of 21): Enhancing training for new employees on dealing with violent incidents; consistently increase the staff wherever it needed; hospital leaders emphasizing the reporting of minor violent incidents; enhancing the capacity of the medical dispute team; and promoting the outstanding HWs for a better public recognition. (b) Improving work environment (6 out of 21): Continuing to deploy volunteers to clearly guide the medical process; minimizing blind spots in equipment monitoring; and increasing security patrols in key areas to timely intervene the possibility of WPV. (c) Enhancing individuals’ coping abilities (6 out of 21): Improving professional capabilities for medical service; increasing personal understanding of WPV prevention and emergency handling; and strengthening doctor–patient communication with an emphasis on empathy. (d) Emphasizing post-incident support (3 out of 21): Developing a reporting, recording, and notifying mechanism for violence incidents; and adopting a blame-free environment toward employees who suffered violence with care and support measures.

Among all the interviewees, 18 expressed their plans for individual improvement in dealing with WPV in the future: (a) Enhancing personal coping abilities (9 out of 21): Actively participating in training on handling violence incidents; increasing awareness and capability for self-protection; improving professional skills to reduce practice mistake; and familiarizing themselves with the reporting procedure of WPV. (b) Strengthening patient-centered care (7 out of 21): Improving communication skill; treating patients sincerely; expressing empathy with patients; and demonstrating the full engagement of HWs to patients. (c) Seeking external support (7 out of 21): Sharing concerns with colleagues; reporting to the supervisor; seeking assistance from the security department; reporting to the police if necessary; reminding colleagues when communicating with difficult patients or their families; and seeking collective support when facing challenging situations.

#### 3.2.6. Intervention of WPV among Administrative and Security Staff

In the unstructured interview, the five members of administrative and security staff fully echoed the information about the intervention of WPV in the Hospital, including the framework of policies, organizational changes, regular training, security system, cultural fostering, patient-centered service, and the harmonization of the doctor–patient relationship. They confirmed the achievements of WPV prevention and control in the Hospital as a whole, and expressed their confidence in intervention in the near future, as well as discussed the challenges going forward.

## 4. Discussion

### 4.1. A Decrease in WPV under Intervention and Prevention Measures, Except for Physical Violence and Sexual Harassment

Governments, employers, health workers, and stakeholders reach a consensus that the incidence of WPV in the health sector is potentially preventable rather than accepted as random acts that are a part of work life. In China, several pilot hospitals have conducted a risk assessment of WPV after introducing the HealthWISE methodology. Comparing with the survey conducted by the Team utilizing the same questionnaire and methodology in a Grade 2A hospital in 2018, the incidence rates of WPV in the same year in this study is relatively higher (60.90% vs. 48.47%) [33]. Additionally, a national cross-sectional survey was conducted in 31 provinces/autonomous regions/municipalities across China, indicating that 90.40%, 51.45%, and 90.00% reported exposure to any type of WPV, physical or nonphysical violence [34]. The Violence Study of Healthcare Workers and Systems—a global survey shows that nearly 55% of HWs reported having experienced firsthand violence, and 16% reported violence against their colleagues [35]. Moreover, a systematic review and meta-analysis of WPV against healthcare workers during the COVID-19 pandemic revealed that WPV (40–47%), physical violence (12–23%), and verbal violence (45–58%) increased from the mid-pandemic to late-pandemic period [36]. Under the (technically guided by the WHO-ILO-China) HealthWISE team, the hospital created a violence-free and human-centered workplace culture. Our study strongly demonstrates that the incidence rate of WPV decreased from 60.90% to 34.44%, and the incidence rate of psychological violence decreased from 60.90% to 33.89% significantly after integrating the anti-violence measures to the OSHMS in the pilot hospital. Furthermore, for psychological violence, the incidence rate of verbal abuse, bullying/mobbing, and ethnic discrimination dropped significantly between this survey and the baseline survey. We also compared the recognition of the perceived 12 anti-violence measures against WPV that were listed on the Questionnaire by participants. It indicated that the percentage of restricting public access, patient screening, patient protocol, changing shifts or rotas, restricting the exchange of money in the workplace, special equipment or clothing, check-in procedures for staff, and increasing staff numbers have increased significantly in endorsement between this survey and the baseline survey. Combined with the decrease in the incidence rate of WPV, this study confirms the well-recognized finding that tackling violence at work by adopting preventive strategies and early intervention is the most effective way to contain and defuse such behavior.

Nevertheless, statistically significant decreases in the incidence rate of physical violence and sexual harassment between this survey and the baseline survey were not found. According to the results of binary logistic regression, HWs in night work were more likely to be exposed to physical violence. Most of the perpetrators of physical violence were patients themselves or their relatives. In line with previous studies, the perpetrator in physical violence was the patients and patient’s family [37,38,39]. Violence by the patient’s family can be related to their concerns about the patient’s condition, unfulfilled expectations, and inappropriate psychological conditions. Similarly, the interviewees in this study also hold the view that the occurrence of physical violence is attributed to unmet treatment expectation, perpetrators’ personal factors (drugs or alcohol abuse, mental disorder, and excessive anxiety as the parents of child patients), etc. Several psychosocial risks, e.g., staffing, workload, unmet treatment outcomes, and working alone, could be improved by working conditions and arrangements, work organization, and human resource management, which is importantly mentioned in ILO C190 and R206.

Sexual harassment in the workplace is characterized by repetitive and unwelcome sexual behaviors, encompassing verbal, physical, psychological, and visual forms [40]. The proportion of perpetrators who commit sexual harassment is more apparent for internal violence. The leading causes could be summarized as sexual harassment being viewed as a concealed, insignificant problem in traditional cultural perceptions. The gender power imbalance in the health setting exacerbates the occurrence of sexual harassment in internal violence. This hospital has not established an unimpeded channel and a victim-protective management protocol. Moreover, the lack of definition and punishment of sexual harassment in labour protection legislation makes it difficult for victims to obtain adequate protection and help.

Internal violence occurs between workers (from co-workers, managers/supervisors, etc.). External violence (third-party violence and harassment) is that which takes place between workers (and managers and supervisors) and any other person present at the workplace. Our findings were consistent with previous studies [41] that reported that the number of perpetrators of external violence was higher than that of internal violence. However, current organizational commitment, and OSHMS and training protocols more often focus on external violence in this hospital rather than internal violence; therefore, internal violence remains unaddressed. It should be emphasized that internal violence should be reduced due to the strong association with a positive psychosocial safety climate and work organization.

### 4.2. Effective Intervention Measures Have Been Taken to Protect HWs from Violence since HealthWISE Tool was Introduced in This Pilot Hospital

The findings of this interview were coherent with the previous study on the long-term mechanism building of occupational health in the Hospital, especially during the COVID-19 pandemic, for combatting WPV [42,43]. Except for the 12 measures provided by the Questionnaire which widely recognized comprehensive measures to address WPV, the following aspect is a summary of the excellent experiences in this Hospital that could be shared with other pilot hospitals and beyond:

Firstly, the creation of a human-centered workplace culture that values equality of opportunity, tolerance, safety and dignity, non-discrimination, and cooperation should take precedence. The Hospital integrated the vision of “zero tolerance for occupational hazards (including WPV)” into the core value of “Guard for Lives” [44], which demonstrates that leadership and staff share a common vision and goals in this Hospital. The hospital leadership has released a clear policy statement emphasizing the significance of the battle against WPV. Taking into account the actual needs of patients, the hospital has taken a number of humane care measures, including providing optimized medical procedures, patient-centered care, reducing the cost of treatment for patients with financial difficulties, setting up a long-term health care mechanism, and creating a cultural publicity bulletin [45]. Apart from promoting positive doctor–patient interactions, these measures also contribute to a minor reduction in the incidence of WPV.

Secondly, the Hospital has implemented organizational interventions in light of specific situations: (1) providing continuous or periodical training in identifying WPV risk factors, resolving conflicts, and managing assaults against all HWs, supervisors, and managers; (2) making improvements in working time, work pace, and management style; (3) providing timely information and humanistic care to patients and their relatives to increase trust between doctors and patients; (4) recording perpetrators’ rosters to help HWs to be as alert as possible and obtain assistance from guards and patrols in case of violent situations; and (5) providing family-friendly measures such as summer kindergarten for HWs’ children; warm accommodation; and adequate assistance to HWs with family difficulties.

Thirdly, the Hospital has implemented environmental intervention to make the physical environment safer: (1) setting up security devices such as metal detectors, surveillance cameras, and one-click alarm devices; (2) assigning adequate guards and patrols in outpatient and ward departments, and setting up 24 h security positions, especially during the COVID-19 pandemic; (3) having separate pathways for HWs and patients, and restricting public access during the COVID-19 pandemic; (4) optimizing the service delivery procedure to reduce waiting time, and providing reading materials, television, and toys for children in waiting area; and (5) providing comfortable environments such as adequate lighting, humidity, temperature, and ventilation.

### 4.3. Stepwise Recommendations for Combatting WPV at the Hospital Level

According the Framework Guidelines for Addressing WPV in the Health Sector, we suggest the key issues as follows:

(a) Pay attention to the prevention and control of physical violence and sexual harassment. The employer has a responsibility to address all forms of internal violence and harassment through primary interventions, with an emphasis on sexual harassment. Internal WPV takes place between workers, including managers and supervisors [46,47]. We suggest that physical violence and sexual harassment must be included in the following risk analysis and evaluation during the “P→D→C” cycle, with particular attention being given to the uniqueness of each workplace situation [48]. Several approaches were recommended as preventive measures to combat sexual harassment, such as enhanced efforts in the promotion of equal rights for women and men, the redistribution of working hours, promoting collective bargaining, and adopting internal procedures for reporting incidents [49].

(b) It is vital that we strengthen the trade unions’ capacity in the Hospital to tackle WPV. According to the Labour Contract Law of the People’s Republic of China and the Trade Union Law of the People’s Republic of China, the trade union has the responsibility of assisting the employer to in ensuring labour safety, health, and social insurance [50,51]. The involvement of trade unions, workers, and all relevant stakeholders can greatly contribute to generating awareness and sensitivity on the issue of WPV [52]. As one of the compulsory affairs, the trade union representatives should negotiate the development of policies to tackle violence with management and the OSHMS committees and provide full training and training updates to HWs who may be at risk of violence.

(c) Establish reporting, recording, and notification systems to assist with WPV identification and elimination. Under-reporting hinders violence prevention efforts in two ways: first, under-reporting results in an underestimation of the true extent of the problem; and, second, without the knowledge of the full spectrum of violent events to which workers were exposed, prevention efforts can only be designed to affect limited aspects of the problem [53]. According to the literature analysis, only 19% of violent events were reported to official reporting systems [54]. To effectively respond to WPV, the Hospital should replace oral reporting procedures with standardized operating procedures and systems for reporting, recording, and notifying violence cases while eliminating barriers to reporting in a blame-free environment.

(d) Taking immediate holistic supportive actions during and after violent incidents. WPV can have a serious negative impact on HWs and even lead to changes from their social relations to social isolation [55,56]. In our study, the victims of violence experienced a wide range of disturbing reactions such as anxiety, feelings of helplessness, disturbed sleep, obsessive thoughts and images, and feelings of shame. In light of the three prevention stages of public health, the Hospital mainly focused on the primary prevention of WPV. However, secondary prevention (emergency services and medical) and tertiary prevention (rehabilitation and reintegration and attempts to lessen trauma or reduce long-term disability) also need to be highlighted [57]. In the long term, coaching and debriefing are suggested to be the regular approaches, as they will reduce WPV through hands-on training that provides the workforce with the knowledge, awareness, skills, and confidence to manage situations [58,59].

(e) Enhancing general self-efficacy may aid in the psychological recovery of victims of WPV. It is recommended that hospital management take relevant measures to increase health workers’ GSE and enhance adaptation and regulation in response to WPV [60]. Furthermore, a positive work environment is critical in preventing WPV and, ultimately, improving work attitudes [61]. WPV is not only a predictor of burnout but, in turn, exacerbates other consequences of WPV [62]. Therefore, management should establish a positive work environment in enhancing post-violence support and management, helping health workers who have experienced WPV alleviate adverse outcomes such as burnout.

### 4.4. Stepwise Recommendations for Combatting WPV at the National Level

The statutory and regulatory provisions that combat WPV include criminal and civil law, occupational safety and health legislation, workers’ rehabilitation and compensation statutes, and environmental and labour laws. Regarding national-level actions, it would be worth referring to the legal obligations in some countries of employers to prevent violence and harassment and carry out a risk assessment as part of occupational safety and health laws, for instance, the Work Health and Safety Act 2011 in Australia [63]; the Act of 4 August 1996 on the well-being of workers in the performance of their work in Belgium [64]; Ontario’s Occupational Health and Safety Act in Canada [65]; the Danish Working Environment Act [66]; the Working Conditions Act in Netherlands [67]; and the Official Gazette n. 224, in Law 113/2020 [68]. It is worth pointing out that, on 19 April 2021, the House of Representatives of the United States passed the ”Workplace Violence Prevention for Health Care and Social Service Workers Act”, which is a specific legislation for combatting WPV in health settings and social service sections [69].

National policy development and practice on anti-violence in the workplace for HWs has been introduced for many years. The Law on Basic Health Care and Health Promotion of China was the first fundamental law in the health sector that stipulated that those who disrupted the order of medical institutions, or threatened or endangered the personal safety of HWs, shall face punishment. Soon after, the first-ever Chinese Civil Code, which took effect in 2021, explicitly defined sexual harassment and imposed affirmative duties on employers. Compared with the new progress in international legislation and practice, there is currently no national guideline in China, with hospital measures mainly focused on responding to extreme violence (criminal cases). In addition, there is no clear definition of WPV and harassment (including sexual harassment) [70,71], gender equality, and internal violence among co-workers fully reflected in national laws. Since ten laws and two administrative regulations relevant to the safety and health of HWs from the legal authority of mental health, infectious diseases, and occupational diseases have been adopted, China has the conditions for the ratification of *the Convention No. 190 and Recommendation No. 206* [72]. National occupational health policy development and laws on WPV prevention for HWs should be accelerated to meet the practical needs of health and well-being. We suggest the key issues of legislation by eliminating potential causes and attaching great importance to the adverse consequences: (1) integrating a workplace perspective to address WPV among HWs; (2) improving investment for staff training and human resources development; (3) adopting a blame-free environment; (4) establishing a fair and effective grievance procedure; (5) clarifying the employer liability doctrine for violence and harassment [73]; and (6) developing and strengthening medical insurance system to share the burden of medical risks. Moreover, even if the PTSD caused by violence and harassment was included in the ILO list of occupational diseases in 2010, this disease has not been included in the Categories and List of Occupational Diseases in China, and, thus, evidence-based research and policy advocacy are urgently needed [74].

### 4.5. Implications of This Study

#### 4.5.1. Strong Leadership Commitment Are Prerequisites for Improving the Working Conditions of HWs

The first step in establishing and implementing effective WPV procedures is strong management commitment. The circulation of information, open communication, and guidance can greatly reduce the risk of violence at work [75]. Leaders of the Hospital participated in the HealthWISE TOT workshop from the beginning, and the Occupational Safety and Health Committee played a leading role covering all departments. As a specialized institution in infectious disease, the Hospital consistently emphasizes the OSHMS and integrates violence prevention activities into daily procedures, including environmental, organizational, and individual interventions to address WPV. The leadership also worked towards establishing and maintaining a culture of zero tolerance to violence, as well as work systems and environments that enable, facilitate, and support the zero-tolerance response. In the unstructured interview, the administration expressed its willingness to negotiate between management and trade union representatives, as well as involve safety representatives and committees at all stages.

#### 4.5.2. The Participation of HWs and Their Representatives Is Crucial in Tackling WPV

The best strategy to tackle WPV is for the leaders of the Hospital and HWs to work together to decide what to do [76]. The participation of HWs and their representatives is crucial both in identifying the problem and in implementing solutions, especially tackling discrimination, harassment, and violence in the workplace, which manifested in the following aspects: most of the interviewees had an accurate understanding of the definition of WPV; the interviewees expressed their willingness to be involved in identifying risk factors and costs of violence at work; and HWs have transformed the knowledge of WPV into active actions successfully.

#### 4.5.3. External Technical Support Facilitating the Capacity Building of the Hospital

Since 2013, with the application of the national occupational health standards, the HealthWISE project had successfully engaged with all the top seven medical colleges in China and 130 hospitals across provinces, leading to a multi-sectoral, multi-disciplinary network with both national and international resources. The project worked closely with the Chinese national and local governmental organizations, trade unions, pilot hospitals and professional organizations, and NGOs, as well as international organizations (the ILO and the WHO). As one of the 130 pilot hospitals, the national program team technically supported the Hospital during the progression from training to implementation via the comprehensive HealthWISE approach since 2015. The Model of Hospital Initiative on Systematic Occupational Health (HISOH Model) has been applied soon afterwards [77]. Besides providing courses, teaching materials, and international research progress on occupational health protection for HWs, systematic approaches toward the upgraded health and well-being of HWs were also sustained, guided by the Team.

### 4.6. Strengths and Limitations

To the best of our knowledge, this is the first investigation to verify the effectiveness of intervention measures against WPV under OSHMS from an international perspective in a Chinese hospital. Compared with the limited data in the previous study, we added sufficient evidence that is hugely useful in ensuring a better focus on occupational safety and health responses to violence and harassment. Moreover, we formulated a further strategy for the piloting practice of the hospital and national laws/policies based on the findings.

Nevertheless, there are some limitations to consider. First, since we investigated the occurrence of violence in the past year, the self-assessment could also have been affected by recall bias. Second, despite the wide distribution of the questionnaire, we reached a comparatively small sample size at the baseline survey, which might lead to selection bias. Especially, HWs who had suffered from WPV may have yet to participate in the survey, which, in turn, could have resulted in an underestimation of the frequency of WPV. Third, the purposive selection of interviewees to share information may have biased the responses. Fourth, both a cross-sectional study and qualitative interview cannot infer causal relationships among the research items, and the persuadability needs to be improved. Moreover, the generalization of the findings to other locations in China needs to be made with caution. We suggest the need for a future study by the Team and other researchers to validate these findings over a wider geography of China, incorporating the perspective of patients and their families on WPV in health settings, and exploring effective measures to reduce the exposure among HWs to WPV.

## 5. Conclusions

This study demonstrates a significant decrease in WPV, and psychological violence, as well as verbal abuse, bullying/mobbing, and ethnic discrimination after implementing the comprehensive OSHMS. The unstructured interview echoed information about the intervention of WPV in the Hospital with complementary inputs. HWs had an in-depth understanding of WPV, which has multifaceted consequences in the context of complex risk factors. Comprehensive measures have been taken for WPV intervention before and after COVID-19, including HealthWISE and national standards.

Going forward, recommendations were provided in the next steps for improving measures against WPV, including paying attention to the occurrence of physical violence and sexual harassment; strengthening active participation of trade union and fostering social dialogue among stakeholders in the Hospital; establishing reporting, recording, and notification systems of WPV; taking immediate holistic, and supportive actions during and after violent incidents; and formulating WPV prevention and control programs, including full participation, specific goals, time schedules, etc.

## Figures and Tables

**Figure 1 healthcare-12-01911-f001:**
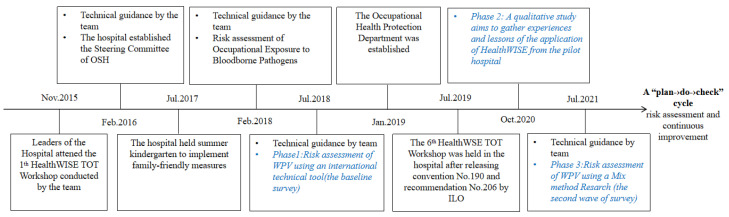
The key milestones of establishing the OSHMS and risk assessment of WPV in the hospital.

**Figure 2 healthcare-12-01911-f002:**
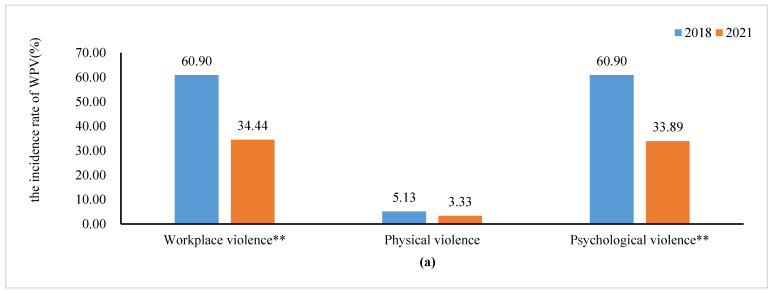
(**a**) Comparison of the incidence rate of various types of WPV between the two waves of survey. (**b**) Comparison of the incidence rate of various types of psychological violence between the two waves of survey. Annotations: ** indicates *p* < 0.01; * indicates *p* < 0.05.

**Figure 3 healthcare-12-01911-f003:**
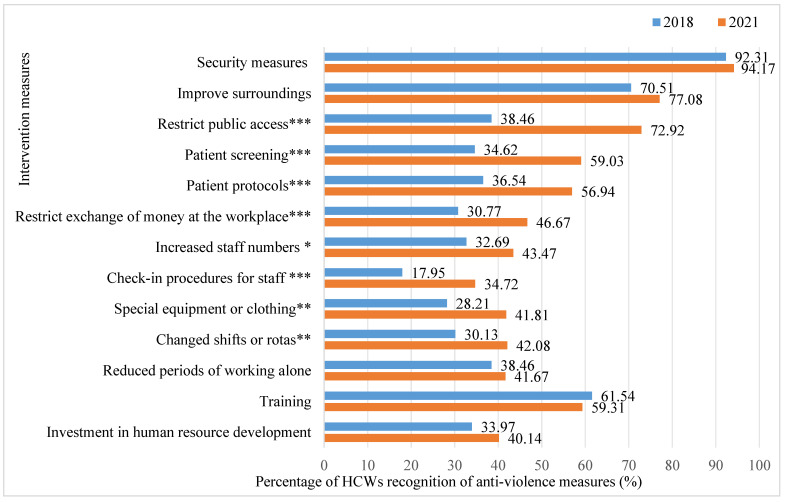
Comparison of the evaluation of perceived anti-violence measures in the workplace among respondents. Annotations: *** indicates *p* < 0.001; ** indicates *p* < 0.01; * indicates *p* < 0.05.

**Table 1 healthcare-12-01911-t001:** Binary logistic regression of different forms of WPV among 720 HWs in 2021.

	Variable	*OR*	95% *CI*
Physical violence			Reference
	Night work *	3.43	1.01 to 11.63
Verbal abuse			
	Professional title		
	Junior or/and below	1.00	Reference
	Middle *	0.60	0.37 to 0.96
	Senior	1.00	0.58 to 1.87
	Have direct physical contact/interaction with patients ***	6.33	2.66 to 15.05
	Worry about WPV ***	2.87	1.68 to 4.89
Bullying/Mobbing			
	Professional title		
	Junior or/and below	1.00	Reference
	Middle *	2.15	1.14 to 4.07
	Senior ***	3.09	1.53 to 6.21
	Department		
	Technical support and administration department	1.00	Reference
	Outpatient and emergency ***	2.79	1.31 to 5.98
	Ward and other	1.35	0.67 to 2.76
	Worry about WPV *	2.96	1.02 to 8.30

Annotations: *** indicates *p* < 0.001; * indicates *p* < 0.05.

**Table 2 healthcare-12-01911-t002:** The occurrence of WPV among 720 HWs in 2021.

	Physical Violence (N = 24)		Verbal Abuse (N = 235)		Bullying/Mobbing (N = 61)		Sexual Harassment (N = 16)		Ethnic Discrimination (N = 9)	
	n	%	n	%	n	%	n	%	n	%
Perpetrator										
Patient/client	10	41.67	104	44.26	22	36.07	9	56.25	6	66.67
Relatives of patient/client	9	37.50	105	44.68	31	50.82	2	12.50	1	11.11
External colleague/worker	0	0.00	0	0.00	0	0.00	0	0.00	1	11.11
General public	1	4.17	5	2.13	1	1.64	2	12.50	0	0.00
Staff member	0	0.00	4	1.70	3	4.92	2	12.50	0	0.00
Management/supervisor	0	0.00	6	2.55	2	3.28	1	6.25	0	0.00
Other	4	16.76	11	4.68	2	3.28	0	0.00	1	11.11
Location										
Hospital	19	79.17	214	91.06	57	93.44	12	75.00	7	77.78
Patients’ home	1	4.17	14	5.96	1	1.64	1	6.25	1	11.11
Outside (on way to work/health visit/home)	2	8.33	6	2.55	2	3.28	1	12.50	0	0.00
Other	2	8.33	1	0.43	1	1.64	2	6.25	1	11.11

**Table 3 healthcare-12-01911-t003:** Overview and description of HWs included by gender and occupation between the two waves of survey.

Characteristic	Baseline Survey (N = 156)		This Survey (N = 720)		χ^2^	*p*
	n	%	n	%		
Gender					6.026	0.01
Male	26	16.67	187	25.97		
Female	130	83.33	533	74.03		
Occupation					1.195	0.274
Technical support and administration	49	31.41	195	27.08		
Frontline HWs (doctor and nurse)	107	68.59	525	72.92		

## Data Availability

All data generated or analyzed during this study are included in this published article and its Appendix A.

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
