# Peer review of "Characteristics and Trends of Workplace Violence towards Frontline Health Workers under Comprehensive Interventions in a Chinese Infectious Disease Hospital"

_healthcare, 2024, doi:10.3390/healthcare12191911_

Round 1

Reviewer 1 Report

Comments and Suggestions for Authors

First of all, I congratulate the authors on the study titled “Characteristics and trends of workplace violence toward frontline health workers under comprehensive interventions in a Chinese infectious disease hospital: A mixed-methods study.” It is a highly relevant topic and should be considered for publication. However, the scope is too broad, which is reflected in the objectives defined for the study. I therefore suggest that the authors consider the provided suggestions and reorganize the structure of the introduction and methods before publication.

  1. Abstract:

    • In the abstract, acronyms should be spelled out the first time they appear (e.g., WPV – “workplace violence”).
    • The abstract should be adjusted according to the comments below.
  2. Introduction:

    • The acronym OSHMS is not spelled out, at least not the first time it appears.
    • From line 76, the authors mention: “Based on the available literature, none of the investigations used a mixed-methods study to verify the effectiveness of measures against WPV under ILO/WHO HealthWISE in China.” However, this is not the only innovation of the present study. The referenced studies suggest investigations in very restricted contexts such as “pregnancy termination,” etc. It should therefore be clear what this study adds to previous ones, including from an international perspective. I would like to see this idea in the introduction.
    • Overall, I think the introduction contains important ideas about standards/guidelines/recommendations on workplace violence (WPV) in health settings. However, the authors lose focus somewhat when trying to explain the methodological framework of the study in the introduction (e.g., “A cross-sectional survey was conducted using a questionnaire developed by the ILO and the WHO in July 2018…”). Much of this explanation should be framed in the methods section, not in the introduction. I suggest reorganizing these sections based on this issue.
    • The pandemic context is addressed (e.g., “practice during the COVID-19 pandemic…”), but what is its impact? How does it change the course of the investigation?
    • The diagram included in the introduction (which, in my opinion, should appear in the methods) clarifies the stages of data collection, but this is not as clear in the text. Essentially, actions take place between collections that can have an impact (e.g., 2018-2021) on the perceived violence in the workplace. In my understanding, we also want to assess the impact of these measures. However, in reality, they began before the study (e.g., with a first workshop). I suggest that the setting of the study provide a more detailed explanation and remove this information from the introduction.
    • There is a need to reformulate the objectives. See suggestions below.
  3. Materials and Methods:

    • See previous comments on the introduction. This is the section where the methodological framework of the study and the different phases of prior recruitment should be made clear.
    • However, from the perspective of the description of the measures and the analyses carried out, I think everything is clear and objective.
  4. Results:

    • The analyses are well-conducted. However, the alignment between the objectives and results should be clearer. In my opinion, these results address the following objectives: (i) Explore the relationship between sociodemographic and work-related factors and the perceived levels of exposure to workplace violence (e.g., Physical violence, Verbal abuse, Bullying/Mobbing, Sexual harassment, and Ethnic discrimination). (ii) Compare the perceived levels of exposure to workplace violence before and after the intensification of anti-violence measures in the hospital context. (iii) Describe the perspectives of frontline HWs on exposure to workplace violence, as well as the relevance of the preventive measures adopted in the hospital context.
  5. Before providing feedback on the discussion, I believe the previous sections should be reviewed and adjusted. Overall, the study is very relevant and incorporates data that could be published. However, the objectives and methodology need to be more clearly explained. Stating that we will compare two waves of data collection is not enough. This would only suffice if the focus were to longitudinally analyze changes in violence before and after the COVID-19 pandemic, but that would already be a different objective. In fact, preventive actions took place that are reflected in the results of this evaluation. So, where are we heading, and what do we want to highlight in this article?

  6. References:

    • The numbering of the references is incorrect, considering the journal's guidelines (e.g., replace i) with [1]; ii) with [2], and so on).

Author Response

Response to Reviewer 1 Comments

1. Summary

Thanks very much indeed for your constructive comments concerning our manuscript. 

2. Questions for General Evaluation

Reviewer’s Evaluation

Response and Revisions

Does the introduction provide sufficient background and include all relevant references?

Must be improved

Improved

Is the research design appropriate?

Can be improved

Improved

Are the methods adequately described?

Must be improved

Improved

Are the results clearly presented?

Can be improved

Improved

Are the conclusions supported by the results?

Yes

3. Point-by-point response to Comments and Suggestions for Authors

Comments 1: First of all, I congratulate the authors on the study titled “Characteristics and trends of workplace violence toward frontline health workers under comprehensive interventions in a Chinese infectious disease hospital: A mixed-methods study.”It is a highly relevant topic and should be considered for publication. However, the scope is too broad, which is reflected in the objectives defined for the study. I therefore suggest that the authors consider the provided suggestions and reorganize the structure of the introduction and methods before publication.

Response 1: Thanks very much indeed for all of your patience and kind comments on our manuscript. Therefore, we have reorganized and modified the structure of the introduction and methods. More detailed revisions can be found in the revised manuscript and the following comments.

Comments 2: [Abstract] In the abstract, acronyms should be spelled out the first time they appear (e.g., WPV – “workplace violence”).

Response 2: Agree. We have modified ”WPV” into” workplace violence (WPV)” ,modified ”HWs” into “Health workers(HWs)”,and modified ”OSHMS” into “occupational safety and health management system(OSHMS)” to emphasize this point.

Comments 3: [Abstract] The abstract should be adjusted according to the comments below.

Response 3: Agree. We have modified and reorganized the abstract.

Comments 4: [Introduction] The acronym OSHMS is not spelled out, at least not the first time it appears.

Response 4: Agree. We have modified ”OSHMS” into “occupational safety and health management system (OSHMS)” in the Introduction section to emphasize this point.

Comments 5: [Introduction] From line 76, the authors mention: “Based on the available literature, none of the investigations used a mixed-methods study to verify the effectiveness of measures against WPV under ILO/WHO HealthWISE in China.” However, this is not the only innovation of the present study. The referenced studies suggest investigations in very restricted contexts such as “pregnancy termination,”etc. It should therefore be clear what this study adds to previous ones, including from an international perspective. I would like to see this idea in the introduction.

Response 5: Agree. We have modified the innovation of the present study as follows: However, it's important to note that the existing body of research is predominantly rooted in the Chinese cultural context, which may not fully capture the characteristics of WPV from an international perspective. Moreover, there is a conspicuous absence of mixed-methods studies that could provide a more comprehensive assessment of the efficacy of intervention strategies against WPV within the framework of OSHMS in Chinese hospitals, particularly when viewed through an international standpoint. This change can be found on page 1 and lines 75~81 of the revised manuscript.

Comments 6: [Introduction] Overall, I think the introduction contains important ideas about standards/guidelines/recommendations on workplace violence (WPV) in health settings. However, the authors lose focus somewhat when trying to explain the methodological framework of the study in the introduction (e.g., “A cross-sectional survey was conducted using a questionnaire developed by the ILO and the WHO in July 2018…”). Much of this explanation should be framed in the methods section, not in the introduction. I suggest reorganizing these sections based on this issue.

Response 6: Agree. Instead of displaying a methodological framework in the introduction section, we introduce the study conception as ”Besides several runs of technical guidance( field visits, remote discussions, and summaries for improvement) by the Team, three phases of research were conducted in the hospital to verify whether systematically implementing the OSHMS improved occupational health protection among HWs.” Then, we have moved the methodological framework from the introduction to the Methods section to emphasize this point.

Comments 7: [Introduction] The pandemic context is addressed (e.g., “practice during the COVID-19 pandemic…”), but what is its impact? How does it change the course of the investigation?

Response 7: Agree. As of January 20, 2020, the first official confirmed cases of coronavirus infection among HWs in China; however, the protection of frontline HWs in all settings was only implemented on January 24 at the national level. Until the middle of March 2020, at least 12 proactive policies were issued and implemented for HWs by the ministries and commissions of the Chinese government. In this paper, the second wave of surveys regarding risk assessment of WPV was conducted in July 2021, which was a relatively stable period of  COVID-19. Therefore, the pandemic context did not change the course of the investigation.

We added that “Risk assessment of occupational hazards and improvement of the OSHMS are continuous and interactive over time. The infectious disease hospitals are high-risk workplaces for Occupational Bloodborne Pathogen Exposure(OBPE) and the occupational exposure status of OBPE was first investigated by the Team. In terms of psychosocial hazards, there is a lack of research on WPV in health settings from the perspective of occupational health protection. Considering the accessibility of participants, we conducted two rounds of surveys in 2018 and 2021. It is worth pointing out, the second wave of surveys regarding risk assessment of WPV was conducted in July 2021, which was a relatively stable period of COVID-19. ” in the Introduction section. This change can be found on page 3 and lines 112~120 of the revised manuscript.

Comments 8: [Introduction] The diagram included in the introduction (which, in my opinion, should appear in the methods) clarifies the stages of data collection, but this is not as clear in the text. Essentially, actions take place between collections that can have an impact (e.g., 2018-2021) on the perceived violence in the workplace. In my understanding, we also want to assess the impact of these measures. However, in reality, they began before the study (e.g., with a first workshop). I suggest that the setting of the study provide a more detailed explanation and remove this information from the introduction.

Response 8: Agree. Actually, risk assessment of occupational hazards and improvement of the OSHMS are continuous and interactive over time. The infectious disease hospitals are high-risk workplaces for Occupational Bloodborne Pathogen Exposure(OBPE); the occupational exposure status of OBPE was first investigated by the Team. In terms of psychosocial hazards, there is a lack of research on workplace violence in health settings from the perspective of occupational health protection. Considering the accessibility of participants, we conducted two rounds of surveys in 2018 and 2021. A more detailed explanation was added in the introduction section,which can be found on page 3 and lines 110~118. Also, we have removed Figure1 from the introduction to the Methods section to emphasize this point.

Comments 9: [Introduction]There is a need to reformulate the objectives. See suggestions below.

Response 9: Agree. We have modified the objectives as ”This study aims to analyze the characteristics and trends of WPV between the two waves of the survey: (i) Explore the relationship between sociodemographic and work-related factors and the perceived levels of exposure to workplace violence (e.g., Physical violence, Verbal abuse, Bullying/Mobbing, Sexual harassment, and Ethnic discrimination). (ii) Compare the perceived levels of exposure to workplace violence before and after the intensification of anti-violence measures in the hospital context. (iii) Describe the perspectives of frontline HWs on exposure to workplace violence, as well as the relevance of the preventive measures adopted in the hospital context ” to emphasize this point. This change can be found on page 3 and lines 121~128 of the revised manuscript.

Comments 10: [Materials and Methods] See previous comments on the introduction. This is the section where the methodological framework of the study and the different phases of prior recruitment should be made clear. However, from the perspective of the description of the measures and the analyses carried out, I think everything is clear and objective.

Response 10: Agree. We have revised and moved the methodological framework from the introduction to the Methods section and revised the different phases of prior recruitment of participants as follows:”In June 2018, valid data from 156 respondents via mobile phone were collected. Participation in the risk assessment of WPV is gradually improved under the systematic implementation of OSHMS. In 2021, the number of HWs who met the inclusion criteria was 820. The data management platform showed that 728 respondents who met the inclusion criteria completed the questionnaire, of whom 720 had valid questionnaires (total response rate 88.78%; total valid response rate 87.80%).” in the “2.3.1. Study population” section. This change can be found on page 4 and lines 163~168 of the revised manuscript.

Comments 11: [Results] The analyses are well-conducted. However, the alignment between the objectives and results should be clearer. In my opinion, these results address the following objectives: (i) Explore the relationship between sociodemographic and work-related factors and the perceived levels of exposure to workplace violence (e.g., Physical violence, Verbal abuse, Bullying/Mobbing, Sexual harassment, and Ethnic discrimination). (ii) Compare the perceived levels of exposure to workplace violence before and after the intensification of anti-violence measures in the hospital context. (iii) Describe the perspectives of frontline HWs on exposure to workplace violence, as well as the relevance of the preventive measures adopted in the hospital context.

Response 11: Agree. We have modified the objectives; please see Response 9.

Comments 12: Before providing feedback on the discussion, I believe the previous sections should be reviewed and adjusted. Overall, the study is very relevant and incorporates data that could be published. However, the objectives and methodology need to be more clearly explained. Stating that we will compare two waves of data collection is not enough. This would only suffice if the focus were to longitudinally analyze changes in violence before and after the COVID-19 pandemic, but that would already be a different objective. In fact, preventive actions took place that are reflected in the results of this evaluation. So, where are we heading, and what do we want to highlight in this article? 

Response 12: Agree. We have modified the objectives and methodology section. We want to highlight in this article that the comprehensive intervention Of occupational health protection for HWs is a systematic approach. Risk assessment of occupational hazards and improvement of the OSHMS are continuous and interactive over time. The infectious disease hospitals are high-risk workplaces for Occupational Bloodborne Pathogen Exposure(OBPE); the occupational exposure status of OBPE was first investigated by the Team. In terms of psychosocial hazards, there is a lack of research on workplace violence in health settings from the perspective of occupational health protection.Considering the accessibility of participants, we conducted two rounds of surveys in 2018 and 2021.

Comments 13: [References] The numbering of the references is incorrect, considering the journal's guidelines (e.g., replace i) with [1]; ii) with [2], and so on).

Response 13: Agree. We have modified the numbering of the references in accordance with the journal's guidelines.

4. Response to Comments on the Quality of English Language

Point 1: I am not qualified to assess the quality of English in this paper.

Response 1: We appreciate your attention to the linguistic aspects of our manuscript. In response, we have carefully revised the language, paying attention to sentence structure, vocabulary choices, and tone. We believe that these improvements will make our paper more accessible and engaging for readers.

5. Additional clarifications

None.

We are grateful to Healthcare for the assistance with the manuscript.

Sincerely,

All Authors

Reviewer 2 Report

Comments and Suggestions for Authors

This is a really excellent and highly relevant article, with the results of a very interesting survey on interventions within am OSHMS. This is an area where there is limited data and constructive outcomes of what works in preventing workplace violence. I am impressed with the methodology and the detailed analysis and description of the results, with evidence that will be hugely useful in ensuring a better focus on occupational safety and health responses to violence and harassment. There has been a dearth of data on this issue. 

Some small points: 

1) In referring to ILO C190/R206, it would be relevant to reference the parts that are relevant to prevention and risk assessment (e.g. Article 9 in C190) as these are hugely important reference for legal frameworks, policy and workplace practices. To be up to date, in five years since the adoption of C190, there have been 44 ratifications (August 2024) and more are in the pipeline.

2) The description of causes of physical and psychological violence are extremely useful and interesting.

Many of the factors that are identified are, in fact, psychosocial risks that may be worth noting this terminology in the narrative e.g. staffing, workload, unmet treatment outcomes and working alone. I mention this also because it is an important in ILO C190 (Article 9) and R206 (Paragraph 8).

3) I would suggest making more of a distinction in the article between internal violence (from co-workers, managers etc.) and external violence (third-party violence and harassment), particularly as it is important to use the language from C190 here. Are there differences between the causes of or the types of violence and harassment, including sexual harassment, from third parties, e.g. patients and families, than from co-workers or managers? I note that reference is made to sexual harassment being more common in internal violence from co-workers and managers – did the research identify why this was the case?

4) If relevant, regarding the HealthWise methodology, do the researchers have any views on its utility for other workplaces? Is it a comprehensive and useful tool, or are there suggestions of ways to enhance its usefulness?

5) At the start of the article and on page 15, the researchers could refer to some more up to date studies that have been carried out in recent years on third-party violence and harassment (TPVH), assault at work, sexual harassment. The data referred to is rather out of date, particularly as so much changed during and after COVID. More recent national and international studies from the health sector tend to show higher rates of violence, harassment and sexual harassment.

6) Regarding national level actions (4.4) it would be worth referring to the legal obligations in other countries tht put a legal obligation on employers to prevent violence and harassment and carry out risk assessment as part of occupational safety and health laws, e.g. Australia Belgium, Canada, Denmark, Netherlands and Italy, amongst others.

7) Participation of workers / trade union is really important to this. Could a stronger case perhaps be made about why social dialogue /joint approaches are likely to have good outcomes.  

8) Finally in the conclusions, it would be good to know what are the next steps that need to be taken to further reduce workplace violence and harassment, beyond what has been successfully achieved in this pilot site. 

Author Response

Response to Reviewer 2 Comments

1. Summary

Thanks very much for your constructive comments concerning our manuscript. With respect to this revision, we have used the comments to further strengthen the quality of this manuscript and have made changes accordingly. We hope that the revised manuscripts and our responses will suffice to qualify our manuscript for publication in Healthcare. We respond separately to each comment of reviewers in detail which were shown below.

2. Questions for General Evaluation

Reviewer’s Evaluation

Response and Revisions

Is the work a significant contribution to the field?

Can be improved

Improved

Is the work well organized and comprehensively described?

Yes

Is the work scientifically sound and not misleading?

Yes

Are there appropriate and adequate references to related and previous work?

Yes

Is the English used correct and readable?

Yes

3. Point-by-point response to Comments and Suggestions for Authors

Comments 1: This is a really excellent and highly relevant article, with the results of a very interesting survey on interventions within an OSHMS. This is an area where there is limited data and constructive outcomes of what works in preventing workplace violence. I am impressed with the methodology and the detailed analysis and description of the results, with evidence that will be hugely useful in ensuring a better focus on occupational safety and health responses to violence and harassment. There has been a dearth of data on this issue.

Response 1: Thank you very much indeed for all of your patience and kind comments on our manuscript. 

Comments 2: In referring to ILO C190/R206, it would be relevant to reference the parts that are relevant to prevention and risk assessment (e.g. Article 9 in C190) as these are hugely important reference for legal frameworks, policy and workplace practices. To be up to date, in five years since the adoption of C190, there have been 44 ratifications (August 2024) and more are in the pipeline.

Response 2: Agree. We have modified and updated as follows” In five years since the adoption of C190, 44 ratifications have been made, and more are in the pipeline.” This change can be found on page 1 and lines 52~53 of the revised manuscript.

Comments 3: The description of causes of physical and psychological violence are extremely useful and interesting. Many of the factors that are identified are, in fact, psychosocial risks that may be worth noting this terminology in the narrative e.g. staffing, workload, unmet treatment outcomes and working alone. I mention this also because it is an important in ILO C190 (Article 9) and R206 (Paragraph 8).

Response 3: Agree. We modified and added that “Several psychosocial risks, e.g. staffing, workload, unmet treatment outcomes, and working alone could be improved by working conditions and arrangements, work organization, and human resource management, which is importantly mentioned in ILO C190 (Article 9) and R206 (Paragraph 8).” This change can be found on page 14 and lines 505~509 of the revised manuscript.

Comments 4: I would suggest making more of a distinction in the article between internal violence (from co-workers, managers etc.) and external violence (third-party violence and harassment), particularly as it is important to use the language from C190 here. Are there differences between the causes of or the types of violence and harassment, including sexual harassment, from third parties, e.g. patients and families, than from co-workers or managers? I note that reference is made to sexual harassment being more common in internal violence from co-workers and managers- did the research identify why this was the case?

Response 4: Agree. In the Materials and methods section, firstly, we have added the following ”For the types of perpetrator, patient/client, relatives of patient/client, external colleague/worker, and general public belongs to internal violence while staff member, management/supervisor belongs to external violence. For the above types of workplace violence, the number of perpetrators of external violence was higher than that of internal violence. For internal violence, the proportion of perpetrators who commit sexual harassment is more apparent. Respondents reported having been sexually harassed by staff member (12.50%) or management/supervisor(6.25%) in the last 12 months (Table 2).” Secondly, as shown in Table 2, we modified the displayed order of perpetrator as patient/client, relatives of patient/client, external colleague/worker, general public, staff member, management/supervisor, and other.

In the discussion section, we first added the definition of external and internal violence from Convention No. 190. Secondly, we analyzed the differences between external violence and internal violence and the causes of sexual harassment in internal violence. The supplementary content is as follows:”The proportion of perpetrators who commit sexual harassment is more apparent for internal violence. The leading causes could summarized as sexual harassment being viewed as a concealed, insignificant problem in traditional cultural perceptions. The gender power imbalance in the health setting exacerbates the occurrence of sexual harassment in internal violence. This hospital has not established an unimpeded channel and a victim‐protective management protocol. Besides, the lack of definition and punishment of sexual harassment in labour protection legislation makes it difficult for victims to get adequate protection and help. Internal violence occurs between workers(from co-workers, managers/supervisors, etc.). External violence (third-party violence and harassment) is that which takes place between workers (and managers and supervisors) and any other person present at the workplace. Our findings were consistent with previous studies that reported that the number of perpetrators of external violence was higher than that of internal violence. However, current organizational commitment, OSHMS and training protocols more often focus on external violence in this hospital rather than internal violence; therefore, internal violence remains unaddressed. It should be emphasized that internal violence should be reduced due to the strong association with a positive psychosocial safety climate and work organization.This change can be found on page 14 and lines 512~529 of the revised manuscript.

Comments 5: If relevant, regarding the HealthWise methodology, do the researchers have any views on its utility for other workplaces? Is it a comprehensive and useful tool, or are there suggestions of ways to enhance its usefulness?

Response 5: Agree. Regarding the HealthWISE methodology, it is practical and participatory for improving the quality of health facilities, based on the principles of the ILO’s programme “Work Improvement in Small Enterprises”(WISE). It focuses on achievements, and promotes the application of simple and low-cost solutions, learning by doing and information exchange.Chinese experience and good practices are introduced in the WHO/ILO guide based on capacities for work improvement in the health sector.

We have modified and added the introduction of HealthWISE in the introduction section as follows ”Technical support plays a pivotal role in developing global and national policies. In 2014, WHO and ILO jointly initiated the international technical tool known as“HealthWISE-Work Improvement in Health Services.” HealthWISE is a practical, participatory methodology for improving the quality of health facilities, based on the principles of the ILO’s programme “Work Improvement in Small Enterprises” (WISE). It promotes the application of smart, simple and low-cost solutions by utilizing local resources, which leads to tangible benefits for workers and their employers. HealthWISE is designed to promote learning by-doing, which encourages managers and staff to work together to raise the awareness of occupational safety and health. A green, healthy, family-friendly workplace with high-quality managing equipment and supplies will also be promoted. This, in turn, helps improve health services’ performance and ability to deliver quality care to patients. China has been using HealthWISE to build capacities for work improvement in the health sector by the WHO-ILO-China HealthWISE team (hereafter referred to as the Team). The training prioritized the HealthWISE modules dealing with general control of occupational hazards, musculoskeletal disorders, biological hazards and infection control, plus tackling discrimination, harassment and violence. The adoption of HealthWISE as a sustainable national program in China has been advocated for in more than 260 pilot hospitals. Chinese experience and good practices are introduced in the WHO/ILO guide titled “Caring for those who care: guide for the development and implementation of occupational health and safety programmes for health workers.”This change can be found on page 1 and lines 82~102 of the revised manuscript.

Comments 6: At the start of the article and on page 15, the researchers could refer to some more up to date studies that have been carried out in recent years on third-party violence and harassment (TPVH), assault at work, sexual harassment. The data referred to is rather out of date, particularly as so much changed during and after COVID. More recent national and international studies from the health sector tend to show higher rates of violence, harassment and sexual harassment.

Response 6: Agree. We have modified and updated more recent national and international studies of violence, harassment and sexual harassment at the start of the article (lines 69~79)and on page 15 (lines 468~476) of the revised manuscript.

Comments 7: Regarding national level actions (4.4) it would be worth referring to the legal obligations in other countries tht put a legal obligation on employers to prevent violence and harassment and carry out risk assessment as part of occupational safety and health laws, e.g. Australia Belgium, Canada, Denmark, Netherlands and Italy, amongst others.

Response 7: Agree. We have modified and updated legal obligations at the national level actions as follows”Regarding national-level actions, it would be worth referring to the legal obligations in some countries on employers to prevent violence and harassment and carry out a risk assessment as part of occupational safety and health laws, for instance, the Work Health and Safety Act 2011 in Australia; the Act of 4 August 1996 on well-being of workers in the performance of their work in Belgium; the Ontario’s Occupational Health and Safety Act in Canada; the Danish Working Environment Act; the Working Conditions Act in Netherlands; the Official Gazette n. 224, in Law 113/2020.” This change can be found on page 16 and lines 628~635 of the revised manuscript.

Comments 8: Participation of workers / trade union is really important to this. Could a stronger case perhaps be made about why social dialogue /joint approaches are likely to have good outcomes.

Response 8: Agree. ILO Declaration on Fundamental Principles and Rights at Work and its Follow-up, adopted at the 86th Session of the International Labour Conference (1998) and amended at the 110th Session (2022) shows that social dialogue between employers, workers and their representatives, and with government is a key element in the successful implementation of anti-violence policies and programmes.

In China, the All-China Federation of Trade Unions (ACFTU) is a powerful trade union organization with nearly 300 million members in 2022. It promotes the occupational health promotion of workers through social dialogue and the tripartite consultation mechanism in the prevention and control of WPV, paying attention to protecting the rights and interests of occupational safety and health for women, promoting the implementation of ILO's HEALTHWISE toolkit and ergonomics checkpoints, developing the protection, care and support for HWs during the COVID-19 pandemic. In addition, the corresponding author in this paper is the the labour protection consultant and one of the 30 experts think tank of ACFTU.

Comments 9: Finally in the conclusions, it would be good to know what are the next steps that need to be taken to further reduce workplace violence and harassment, beyond what has been successfully achieved in this pilot site.

Response 9: Agree. We modified and added the recommendations provided in the next steps to further reduce workplace violence and harassment. The supplementary content is as follows:”Going forward, recommendations were provided in the next steps for improving measures against WPV, including paying attention to the occurrence of physical violence and sexual harassment; strengthening active participation of trade union and fostering social dialogue among stakeholders in the Hospital; establishing reporting, recording, and notification systems of WPV, taking immediate holistic, and supportive actions during and after violent incidents, and formulating WPV prevention and control programs, including full participation, specific goals, time schedules, etc.” This change can be found on page 18 and lines 736~743 of the revised manuscript.

4. Response to Comments on the Quality of English Language

Point 1: English language fine. No issues detected.

Response 1: We appreciate your attention to the linguistic aspects of our manuscript.   

5. Additional clarifications

None.

We are grateful to Healthcare for the assistance with the manuscript.

Sincerely,

All Authors

Reviewer 3 Report

Comments and Suggestions for Authors

Dear authors,

the title of the paper is too long - would be better a shorter one.

- the comprehensive intervention would be better described. 2021 was in COVID periode = not a `normal` periode. 

- It would be better to put the long tables in the annex. 

- abbreviations would be better to explain from the first use (not at the end)

- you have too short subchapters, these can be united. 

- every figure must include explications. 

- the paper must be edit

- statistically, the 3.1.4. subchapter is not clear to me, must make a better evaluation

Author Response

Response to Reviewer 3 Comments

1. Summary

Thanks very much for your constructive comments concerning our manuscript. With respect to this revision, we have used the comments to further strengthen the quality of this manuscript and have made changes accordingly. We hope that the revised manuscripts and our responses will suffice to qualify our manuscript for publication in Healthcare. We respond separately to each comment of reviewers in detail which were shown below.

2. Questions for General Evaluation

Reviewer’s Evaluation

Response and Revisions

Does the introduction provide sufficient background and include all relevant references?

Can be improved

Improved

Is the research design appropriate?

Can be improved

Improved

Are the methods adequately described?

Can be improved

Improved

Are the results clearly presented?

Can be improved

Improved

Are the conclusions supported by the results?

Can be improved

Improved

3. Point-by-point response to Comments and Suggestions for Authors

Comments 1: the title of the paper is too long - would be better a shorter one.

Response 1: Thank you for pointing this out. We agree with this comment. Therefore,we have modified the title as ”Characteristics and trends of workplace violence toward frontline health workers under comprehensive interventions in a Chinese infectious disease hospital”.

Comments 2: the comprehensive intervention would be better described. 2021 was in COVID periode = not a `normal` periode. 

Response 2: Agree. It is essential that anti-violence action be carried out in a systematic way,the comprehensive intervention could be described as: (1) developing a human-centred workplace culture in pre-conditions,and a clear policy statement of intent should be issued from the top management in consultation with all stakeholders recognizing the importance of the fight against workplace violence. (2)High priority should be given to organizational intervention:the adequate presence of staff, management style, circulation of information and open communication, changing and improving work practices, job design, and working time. (3)Environmental interventions may include:physical environment,workplace design, safe access, adequate work space, comfortable waiting areas, and alarm systems and surveillance cameras. (4)Individual-focused interventions should be developed by training, assistance and counselling, well-being promotion. (5)After the-event-interventions:response management plans, reporting and recording systems,medical treatment, de-briefing,etc.

Risk assessment of occupational hazards and improvement of the OSHMS are continuous and interactive over time. Considering the accessibility of study setting and participants, we conducted two rounds of surveys in 2018 and 2021. In this paper, the second wave of surveys regarding risk assessment of WPV was conducted in July 2021, which was a relatively stable period of COVID-19.

Comments 3: It would be better to put the long tables in the annex. 

Response 3: Agree. We have modified Table 1 into Supplementary Materials S3, Table 2 into Table 1, Table 3 into Table 2, and Table 4 into Table 3 to emphasize this point.

Comments 4: abbreviations would be better to explain from the first use (not at the end).

Response 4: Agree. We have modified”WPV”into”workplace violence (WPV)”, modified”HWs” into “Health workers (HWs)”, and modified “OSHMS” into “occupational safety and health management system(OSHMS)” to emphasize this point. This change can be found on page 1 of the revised manuscript.

Comments 5: you have too short subchapters, these can be united. 

Response 5: Agree. We have united short subchapters in paragraph 2 in “4.6. Strengths and limitations” to emphasize this point. This change can be found on page 18 and lines 721~724 of the revised manuscript.

Comments 6: every figure must include explications. 

Response 6: Agree. We have modified the ordinate name of Figure 2a to "the incidence rate of WPV(%)" and modified the ordinate name of Figure 2b to "the incidence rate of various types of psychological violence(%)." We also provided the respective explanations in Figure 2a and Figure 2b to emphasize this point. This change can be found on page 9 and lines 282~289 of the revised manuscript.

Comments 7: the paper must be edit.

Response 7: Agree. We have edited the total paper to emphasize this point.

Comments 8: statistically, the 3.1.4. subchapter is not clear to me, must make a better evaluation. 

Response 8: Agree. In the 3.1.4. subchapter, we compared the incidence rate of various types of workplace violence (physical violence, verbal abuse, bullying/mobbing, sexual harassment, and ethnic discrimination) between the two waves of the survey. Firstly, Cochran-Armitage (CA) trend test was used to verify the change in gender and occupation over time. Then, the incidence rate of various types of WPV between the two waves of the survey was compared in Figure 2a and Figure 2b.

We have also added a more detailed explanation, such as ”3.1.4. Changes in frequency of workplace violence between the two waves of survey” ”Table 3. Overview and description of HWs included by gender and occupation between the two waves of survey(2018,2021)” and “Figure 2. Comparison of the incidence rate of various types of WPV between the two waves of survey” to emphasize this point.

4. Response to Comments on the Quality of English Language

Point 1: I am not qualified to assess the quality of English in this paper.

Response 1: We appreciate your attention to the linguistic aspects of our manuscript. In response, we have carefully revised the language, paying attention to sentence structure, vocabulary choices, and tone.

5. Additional clarifications

None.

We are grateful to Healthcare for the assistance with the manuscript.

Sincerely,

All Authors